# A Scoping Review on the Epidemiology of Chronic Low Back Pain among Adults in Sub-Saharan Africa

**DOI:** 10.3390/ijerph19052964

**Published:** 2022-03-03

**Authors:** Morris Kahere, Mbuzeleni Hlongwa, Themba G. Ginindza

**Affiliations:** 1Discipline of Public Health Medicine, School of Nursing and Public Health, University of KwaZulu-Natal, Durban 4041, South Africa; hlongwambu@gmail.com (M.H.); ginindza@ukzn.ac.za (T.G.G.); 2Burden of Disease Research Unit, South African Medical Research Council, Cape Town 7505, South Africa

**Keywords:** chronic low back pain, epidemiology, risk factors, prevalence, comorbidities, disability

## Abstract

Background: The global burden of chronic low back pain (CLBP) is a major concern in public health. Several CLBP epidemiological studies have been conducted in high-income-countries (HICs) with little known in low-and-middle-income-countries (LMICs) due to other competing priorities of communicable diseases. The extrapolation of results of studies from HICs for use in LMICs is difficult due to differences in social norms, healthcare systems, and legislations, yet there is urgent need to address this growing burden. It is against this backdrop that we conducted this review to map the current evidence on the distribution of CLBP in Sub-Saharan Africa (SSA). Methods: A comprehensive literature search was conducted from the following databases: PubMed, Google Scholar, Science Direct databases, World Health Organizations library databases, EMBASE, EBSCOhost by searching the following databases within the platform; academic search complete, CINAHL with full text, health sources: nursing/academic and MEDLINE. The title, abstract and the full text screening phases were performed by two independent reviewers with the third reviewer employed to adjudicate discrepancies. The reference list of all included articles was also searched for eligible articles. This scoping review was reported in accordance with the PRISMA extension for scoping reviews (PRISMA-ScR): checklist and explanation, as well as guided by Arksey and O’Malley’s scoping review framework. A thematic content analysis was used to give a narrative account of the review. Results: The electronic search strategy retrieved 21,189 articles. Title/abstract and full text screening only identified 11 articles, which were included in this review. The prevalence of CLBP among the general population ranged from 18.1% to 28.2% and from 22.2% to 59.1% among LBP patients. The prevalence of occupation based CLBP ranged from 30.1% to 55.5%. Identified risk factors for CLBP are multifactorial and included biomechanical, psychological, socioeconomic and lifestyle factors, with psychosocial factors playing a significant role. Hypertension, diabetes mellitus, peptic ulcer disease were the most common comorbidities identified. CLBP disability was significantly associated with psychosocial factors. The management of CLBP in primary care follows the traditional biomedical paradigm and primarily involves pain medication and inconsistent with guidelines. Conclusions: There are limited epidemiological data on CLBP in SSA, however, this study concluded that the prevalence and risk factors of CLBP in SSA are comparable to reports in HICs. Considering the projected increase in the burden of CLBP in LMICs extensive research effort is needed to close this knowledge gap.

## 1. Background

Low back pain (LBP) is increasingly becoming a major public health concern with an estimated global lifetime prevalence of 70–85% [1]. According to the Global burden of disease (GBD) 2017, the global years lived with disability (YLD) were 42.5 million (95% UI: 30.2–57.2 million) in 1990 and increased by 52.7% to 64.9 million (95% UI: 46.5–87.4 million) in 2017 [2]. In 2019, the global LBP prevalent cases were 568.4 million, with an age-standardized point-prevalence of 6972.5 per 100,000 population, and 223.5 million incidence cases with an age-standardized annual incidence of 2748.9, globally [3]. Low back pain is now the global leading cause of disability and work absenteeism, associated with huge socioeconomic burden and production loss [4]. Globally, approximately 149 million work days are lost annually due to LBP resulting in a considerable amount of production loss [5]. Among people under the age of 45, LBP is the second most common reason to visit a physician after the common cold, the third most cause of surgical procedures and the fifth ranking cause of hospital admissions [6].

Low back pain is defined as pain, muscle tension or stiffness localized below the costal margin and above the inferior gluteal fold with or without pain radiating down to the legs and is classified as being specific or non-specific [7]. Specific LBP is identified by a known/specific pathophysiologic cause such as hernia, infection, osteoporotic fractures, tumors, inflammation or rheumatoid arthritis [8]. Only a small proportion (≈10%) of LBP diagnosed individuals have an identified specific underlying cause [7]. The majority (≈90%) of LBP presentations are non-specific which means the etiology is unknown, and the diagnosis is made based on the exclusion of a specific pathology [9]. Non-specific LBP can be classified further according to the duration of symptoms as acute (<6 weeks), subacute (>6 weeks <3 months) and chronic (3 months) LBP [6]. Non-specific LBP usually resolve within a few weeks with minimum or no intervention, however, in some cases, there will be episodes of recurrent pain and disability requiring targeted, multidisciplinary interventions [10]. Only 10% to 20% of LBP sufferers develop CLBP resulting in episodes of excruciating pain, significant physical disability and activity limitation [5]. Despite its small proportion, CLBP is responsible for the majority of the burden attributed to LBP, globally [11]. The cause of CLBP is still a subject of debate, with several theories having been postulated in previous years trying to describe the etiological mechanism, however, the mosaic of its pathophysiology is difficult to understand [12].

Although there is abundant literature on LBP, evidence of CLBP is limited. CLBP is often described secondarily as either a subheading or just a few sentences in studies investigating LBP or other musculoskeletal conditions [13]. Nevertheless, there is increasing evidence of CLBP HICs, though little is known in LMICs. Despite the recognized global burden of CLBP, it is still regarded as a trivial condition in LMICs, where the current research efforts and funding are directed towards life-threatening communicable diseases associated with high mortality rates such as HIV/AIDS, tuberculosis and the current COVID-19 pandemic [4]. In addition to the burden of these epidemic diseases and other competing priorities, health care and social systems in LMICs are already overstretched and not equipped enough to deal with the increasing burden of CLBP [4,14]. Thus, the burden of CLBP is projected to continue increasing in these contexts [4]. Reviews of LBP studies have been conducted in both HICs and LMICs, including Africa [13,14,15,16,17,18]. However, reviews on CLBP are limited in HIC [13], with none identified in LMICs including Africa. Therefore, it is against this backdrop that we conducted this current review to map the evidence on the distribution of CLBP among adults in SSA.

## 2. Materials and Methods

### 2.1. Scoping Review

A scoping review was adopted as it was deemed appropriate to answer the research question. We conducted a scoping review of grey literature and published peer-reviewed articles to map the available evidence on the prevalence, associated risk factors, disability, comorbidities, and management of CLBP in SSA. This review was guided by the Arksey and O’Malley (2005) methodological framework for scoping reviews [19]. This framework is comprised of the following five steps, (I) Identify the research question, (II) Identify the relevant studies, (III) Study selection, (IV) Charting the data, and (V) Collating, summarizing and reporting data [19]. We performed a methodological quality appraisal of included studies as recommended by Levac et al. [20]. This study was reported in accordance with the MOOSE guidelines for observational studies in epidemiology and the preferred reporting items for systematic reviews and meta-analysis extended for scoping reviews (PRISMA-ScR) checklist and explanation [21]. A protocol for this review was published a priori [22].

### 2.2. Identification of the Research Question

This scoping review sought to answer the research question, “What is the existing evidence on the distribution of CLBP among adults in SSA?”. The following sub-questions were considered:What is the prevalence of CLBP among adults in SSA?What are the risk factors associated with CLBP among adults in SSA?What are the comorbidities associated with CLBP among adults in SSA?What are the factors associated with CLBP disability among adults in SSA?What are the current management practices for CLBP in SSA?

### 2.3. Identification of Relevant Studies

We performed a scoping review which included all study designs of published peer-reviewed articles and grey literature aiming to identify the relevant studies to answer the research question. A comprehensive key word electronic literature search was performed in December 2021 with a 10-year date limit in order to retrieve contemporary data relevant to this review. The following databases were searched: EBSCOhost platform by searching the following databases within the platform: Academic search complete, health source: nursing/academic edition, CINAHL with full text, Embase, PubMed, MEDLINE, Science Direct databases, Google Scholar, and World Health Organization (WHO) library databases and grey literature to retrieve articles that are relevant to this scoping review, guided by the study inclusion and exclusion criteria. An initial search of PubMed and CINAHL was conducted, followed by an analysis of text words contained in the title and abstract, and of the index terms used to describe the article. This informed the development of a search strategy which was tailored for each information source. The reference list of all studies eligible for inclusion were also screened for potential additional studies. Attempts were also made to contact authors of potentially relevant articles in order to obtain further information on this topic, however, this effort did not yield any additional articles. Boolean terms (AND, OR) and Medical Subject Headings (MeSH) terms formed part of our search strategy. The database search was conducted using the following key word terms; “back pain,” “lumbago,” “back ache,” “backache,” “lumbar pain,” “lumbar spine pain,” “sciatica,” “degenerated disk,” “degenerated disc,” “degenerative disk,” “degenerative disc,” “displaced disk,” displace disc,” “prolapse disk,” “prolapsed disc,” “spinal stenosis,” “intervertebral disk,” “intervertebral disc,” “Intervertebral Disc Displacement,” “slipped disc,” “slipped disk,” “herniated disk,” “herniated disc,” “disk prolapse,” “disc prolapse,” “disc herniation,” “disk herniation,” “disk protrusion,” “disc protrusion,” “protruded disc,” “protruded disk,” “degenerative spine,” “spinal stenosis,” “coccyx,” “tailbone pain” and “osteoarthritis”. Sub-Saharan Africa country names, and truncated terms such as ‘East-Africa’ were also used to ensure that articles indexed using SSA country-specific names or regional terms were retrieved.

### 2.4. Study Selection and Eligibility

The initial database search was conducted by the principal researcher working closely with a senior scientist from Cochrane South Africa to retrieve eligible articles. Studies obtained through database searches were exported to the Endnote version 8 reference management software, where all the duplicates were removed using the find duplicates function. Two independent reviewers (MK and HC) then conducted title/abstracts and full article screenings, guided by eligibility criteria for this review. Any disagreements between the two reviewers were adjudicated by the third reviewer (CM) until a consensus was reached. The reference list of those articles that were found eligible for the study was searched for studies which may not have been identified during database search. The Preferred Reporting Items for Systematic Reviews and Meta-Analysis (PRISMA) chart (Figure 1) was used to document the review process [23].

### 2.5. Inclusion Criteria

This review only included studies conducted among adults in SSA and published in English or those published in other languages but with an accessible English version and with a clear definition of CLBP in terms of its anatomical characterization and duration of symptoms and presented evidence on either of the following:Prevalence of CLBP;Risk factors for CLBP;Comorbidities associated with CLBP;Factors associated with CLBP disability;Management of CLBP (how CLBP is managed).

### 2.6. Exclusion Criteria

Studies were excluded if they did not present evidence on any of the characteristics described above set as the criteria for inclusion and those studies conducted outside of the SSA region or published in other language with no identifiable English version. We also excluded from this review, studies that lacked a clear definition of CLB and studies conducted among children or adolescence.

### 2.7. Charting the Data

Data from all the included articles were extracted by two the independent reviewers (MK and HC) using a data extraction form which was developed by the principal researcher and the results are presented in Table 1. The data extraction form was piloted to test the consistency of the data extraction process, and the necessary amendments were made prior to the commencement of the final use of the tool. The following information was extracted from the included studies: author and publication year, country, study setting, design, sample size, population description, age group and main findings. NVivo 12 software was used to organize the extracted data into different themes. The extracted data were then collated and summarized.

### 2.8. Collating and Summarising Findings

The extracted data were continually reviewed to improve the quality of the collated and summarized evidence. A thematic content analysis was used to give a narrative account of the extracted evidence [34].

### 2.9. Methodological Quality Appraisal

We assessed the quality of included studies (Table 2) using a tool adopted from Hoy et al. (2012) which used the following domains to assess the risk of bias: sample frame, sample size estimates, randomization used, likelihood of non-response bias, validity of the study instruments, standardization of data collection, use of human body drawings and if the data were collected directly from the study participants [17]. These domains were weighted and a score of 0.2 was given to the first three (on the list before) because they had a higher chance of causing bias and a score of 0.08 was given to the remaining five elements. A quality of less than 50% was regarded as poor, 50–74% was regarded as good and 75% and above was regarded as of excellent quality. Only one study (9.1%) was of poor quality, while the majority of the studies (82%) were of excellent quality, with one (9.1%) good quality study.

## 3. Results

The electronic search strategy retrieved 21,189 references (Figure 1), which were screened for titles. After application of the automation tools during the database search stage, 20,741 articles were excluded because they did not meet the inclusion criteria. Fourteen duplicates were removed, and 28 articles were excluded for other reasons, leaving 406 articles which were screened for titles. A total of 302 articles were removed at the title screening stage because they formed part of the exclusion criteria, leaving 104 articles which were further screened for abstracts. The abstract screening stage excluded 22 articles which were considered part of the exclusion criteria. The remaining 82 articles underwent full-text screening and 71 as the had no evidence on CLBP among in SSA. Therefore, 11 articles met the inclusion criteria and were included in the quality assessment stage.

### 3.1. Characteristics of Included Studies

We included 11 studies conducted in the SSA region and include the following countries: Nigeria (27.3%, n = 3/11), Ethiopia (27.3%, n = 3/11), South Africa (27.3%, n = 3/11), Ghana (9.1%, n = 1/11) and Cameroon (9.1%, n = 1/11), Table 3. All the studies included in this review were cross-sectional in design. More than a third (36.4%, n = 4) of the included studies were hospital based, four (36.4%) were population based, one (9%) was health facility based, one (9.1%%) was institution-based, one (9.1%%) was a port-based study. There was variation in the study population among the included studies. Three studies (27.3%) were occupational based investigating the following three occupations: teachers, farmers, and truck drivers, while four (36.4%) involved the general population, and four (36.4%) studies investigating LBP patients. Only one study investigated CLBP as the outcome of interest, while most of the studies were investigating LBP and only describing CLBP as either a subheading or in just a few sentences. Based on the thematic content analysis, the following themes emerged: prevalence, risk factors, comorbidities, disability, management practices and cost.

### 3.2. Prevalence of Chronic Low Back Pain

The prevalence of CLBP varied considerably among studies in SSA mainly due to the differences in the study population, study methodology and lifestyle factors. Ninety-one percent (91%, n = 10) of the included studies reported the prevalence of CLBP (Table 4). Based on the findings of this review, the prevalence of CLBP is categorized into three subcategories, CLBP prevalence among the general population, CLBP prevalence among LBP patients and occupational-based CLBP prevalence. The prevalence of CLBP among the general population ranged from 18.1% to 28.2% [27,30,32], and from 22.2% to 59.1% among LBP patients [25,31,33,35]. Occupational-based CLBP prevalence ranged from 30.1% to 55.5% [1,35,36]. The prevalence of CLBP among primary school teachers, farmers and long-distance truck drivers were 30.1% [1], 48.5% [26] and 55.3% [28], respectively. A retrospective cross-sectional hospital-based study among adult LBP patients seen in an orthopedic clinic in Nigeria reported a CLNP prevalence of 59.1% [25]. Beyera et al. conducted a cross-sectional hospital-based study analyzing factors associated with hospital admissions in Ethiopia following presentation for LBP [29]. Beyera et al. reported that, the prevalence of CLBP lasting for 1–5 years and >5 years were 38.7% and 16.2%, respectively [29]. A multicounty study in Ghana investigating age and gender specific burden of chronic musculoskeletal disorders among elderly population reported a 28.2% prevalence of CLBP [27]. Nakua et al. reported that uneducated females had a higher prevalence of CLBP (36.2%) compared to their male counterparts (29.0%). Among professionals with technical skills, the prevalence of CLBP was higher among females (40.8%) compared to males (28.0%) [27]. A cross-sectional health facility-based study conducted in South Africa reported a 26.3% prevalence of CLBP [32]. This finding was slightly higher than the 18.1% which was reported among adults presenting to public hospitals in KwaZulu-Natal in South Africa [30]. Among LBP patients in South Africa, the prevalence of CLBP was reported to be 22.2% [31]. A central African study conducted in Cameroon reported a 41% prevalence of CLBP, with 56% of them having radiculopathy and 3% with CLBP from a specific spinal cause [33].

### 3.3. Risk Factors Associated with Chronic Low Back Pain

After an extensive and thorough search of current literature, we only identified two study which investigated CLBP as an outcome of interest, and two investigating factors associated with disability among CLBP patients. However, based on the evidence retrieved CLBP is a multifactorial condition whose etiology can be predicted by physical/biomechanical [1,28,36], psychological [24], socioeconomic [24,27], individual/lifestyle and demographic factors [27].

Physical or biomechanical risk factors identified in this review can be categorized into high-impact trauma (such as motor vehicle accidents, slip or falls, sports injuries, whole body vibration [28]), repetitive microtrauma (such as occupation biomechanical stress, poor postural ergonomics [26], prolonged standing [1] or sitting [30], repetitive bending and or twisting movements [26]) and heavy or unnatural loading [26] (lifting too heavy, pulling too heavy, pushing too heavy). Kebede et al. reported that prolonged standing was a significant risk factor for CLBP among primary school teachers in Ethiopia [1]. Prolonged bending and long years of service were the two main identified risk factors associated with CLBP among South-West Nigerian farmers [26]. Frequent lifting or carrying heavy objects and perceived improper sitting posture while driving were the main biomechanical risk factors for CLBP among long distance truck drivers in Ethiopia [28]. Sedentary lifestyle, manual work and a stooped sitting posture were the identified physical or biomechanical risk factors for CLBP among the general adult population attending public hospitals for health related services in KwaZulu-Natal, South Africa [30].

The identified psychological risk factors encompassed cognitive, emotional, and behavioral factors [24,33]. A cross-sectional study conducted in Ethiopia found that perceived improper sitting posture while driving, and perceived job stress were the contributing factors for CLBP. Similarly, another cross-sectional study in Cape Town, South Africa reported that CLBP was significantly associated with a high score of psychosocial distress among adults [32]. This was in agreement with what was reported among adults in KwaZulu-Natal, South Africa, that disease conviction, affective disturbance, denial, and fear avoidance beliefs about work were significant risk factors for CLBP [31]. Similar findings were also observed in Nigeria among the rural adult population, that abnormal illness perception, pain intensity, catastrophizing, anxiety, fear avoidance behavior were significant predictors of CLBP and disability [24]. Additionally, severe pain intensity was significantly associated with CLBP and disability in Cameroon [33].

Socioeconomic factors identified as important risk factors for CLBP include lack of formal education (leading to poor health literacy), poverty or a low socioeconomic status and lack of social or family support [24,27]. A study conducted in Ghana found a positive association between socio-economic status and the development of CLBP [27]. A study in Ghana reported that, low educational status was significantly associated with CLB [27]. The prevalence of CLBP was higher among women with primary education compared to those with university degrees [27]. A higher prevalence of CLBP was observed among subsistent farmers (18.9%) compared to traders (17.2), civil servants (17.2%), students (14.8%) and teachers (12.7%) in Nigeria. A cross-sectional study on the biopsychosocial factors associated with CLBP disability in Nigeria found that lack of social support was significantly associated with CLBP and disability [24].

The current review identified several individual or lifestyle factors associated with CLBP among adults which include, excessive or chronic alcohol consumption [29,30], cigarette smoking [28,30], overweight/obesity [30], lack of regular physical exercises [1,28,36]. A study in Ethiopia reported that, the likelihood of CLBP was higher among those who were physically inactive compared to those who exercised regularly [1]. Another cross-sectional study among long distance truck drivers in Ethiopia identified the following lifestyle factors, cigarette smoking and physical inactivity as significant risk factors for CLBP [28]. Similarly, this concurs with a South African based study which identified overweight, lack of regular physical exercises, light and heavy cigarette smokers, occasional and frequent consumption of alcohol significantly associated with CLBP among adults [30]. This agrees with another study in Nigeria which reported that, among a list of health compromising behaviors, only alcohol consumption was significantly associated with hospital LBP admissions [29].

Some demographic factors were found to be associated with CLBP in this review and these include advanced age [28,30,32], the female gender [27,30] and number of pregnancy among females [1,28,33]. Age was significantly associated with CLBP and disability among adults in Cameroon [33]. Similarly, elderly men aged 70 and above were at greater risk of developing CLBP compared to the 50–59 years age group in Ghana [27]. Although several studies have reported a higher prevalence of CLBP among females compared to males, a contradictory finding was observed among peasant farmers in Southwest Nigeria where males (78.4%) had a higher prevalence of CLBP compared to females (21.6%) [26]. Incidence of CLBP was higher among Nigerian females compared to their males counterparts [25]. In South Africa, the female gender was identified as a significant demographic risk factor for CLBP among adults [30]. Hospital admissions due to CLBP was higher among females compared to males in Ethiopia [29]. A female gender stratified analysis in South Africa found that the number of pregnancy was a significant predictor of CLBP [30].

### 3.4. Chronic Low Back Pain Comorbidities and Disability

The most common comorbidities associated with CLBP identified in this review were hypertension, diabetes mellitus and peptic ulcer disease [25]. CLBP accounted for 61.9% of the patients that presented with hypertension. Osteoarthritis of the knee and hip and cervical spondylosis were the top three musculoskeletal comorbidities identified [25]. CLBP was significantly associated with disability and decreased quality of life among Nigerian farmers [26]. Nigerian farmers suffered difficulties in carrying on day-to-day farm work, had sleeping disturbances and were prevented from sexual activities, attending parties, hunting and going to the markets due to CLBP [26]. Similarly, the likelihood of CLBP was seven times higher among teachers who had sleeping disturbances compared to those who had no problems with sleep in Ethiopia [1]. Absenteeism from work was reported by two-thirds (64.3%) of LBP patients in Ethiopia [29].

A cross-sectional study among Nigerian adults with CLBP investigated the biomechanical and psychosocial predictors of CLBP disability in rural Nigeria and found that, psychosocial factors were the most important factors associated with disability [24]. Abnormal illness perception, pain intensity, catastrophizing, fear avoidance beliefs and anxiety were the significant predictors of self-reported disability while performance based disability was significantly predicted by abnormal illness perception, lack of social support, fear avoidance beliefs and the female gender [24]. Illness concern was the most noticeable subscale of illness perceptions predicting self-reported and performance-based disability [24]. Igwesi-Chidobe et al. reported that occupational biomechanical factors did not predict CLBP disability [24]. A similar study in Cameroon reported that the prevalence of CLBP disability was 88.1% among CLBP patients. According to Doualla et al., longer work absence and greater pain intensity moderately correlated with higher scores of CLBP disability [33]. Alcohol consumption and bladder/bowel dysfunction syndrome (BBDS) were significantly associated with CLBP disability [33]. Age and the duration of pain had a weak positive relationship with disability. Expectedly, sleep satisfaction and psychological wellbeing significantly contributed to less disability. Non-alcohol consumers, BBDS and sciatica patients had higher disability scores. Pain intensity, days of work absence, psychological well-being, alcohol consumption and BBDS as independently and significantly associated with disability.

### 3.5. Patterns of CLBP Presentations, Hospital Admissions and Management Strategies

A hospital-based study in Nigeria investigated the etiology and patterns of presentation of LBP among patients [25]. And found that, mechanical CLBP was the most prevalent (82.1%), with a significantly higher incidence among females (53.1%) than males (46.9%). According to this study by Omoke et al., the incidence of LBP increased from the fourth decade, doubled in the fifth decade, reached peak in the seventh decade among farmers, whereas it doubled in the fourth decade and reached peak in the fifth decade among civil servants [25]. The incidence among teachers and petty traders reached peak in the seventh and fifth decade, respectively. Lumbosacral spondylosis in 43.1% of the patients was the leading cause (52.3%) of mechanical LBP, followed by intervertebral disc prolapse that was responsible for pain in 23.4% of the patients. LBP attributed to spondylosis, spondylolisthesis, sacroiliac joint strain and back strain was significantly higher in females than males, *p* < 0.015, whereas disc prolapse was more in males than females. CLBP was accounted for by spondylosis, Potts’s diseases and disc prolapse in 51.7%, 14.5% and 13.4% of the patients, respectively. Insidious onset of LBP was observed in 61.9% of patients. The anatomical characterization of LBP involved one spinal vertebral level in 37.8% of the patients, two spinal vertebral levels in 33.0%, and three or more levels in 10.65% of the patients. L4-L5, L5-S1 and L2-L2 spinal vertebral levels were involved in about 42%, 26.8% and 26.1% of the patients.

Hospital admissions and associated factors following presentation for CLBP was investigated in Ethiopia [29]. The proportion of admissions following presentation for CLBP was 14.4%. The admission rate was higher in females (18.5%) than in males (11.4%). Among those hospitalized, 26.9% were managed by surgical interventions, which was 3.9% of the total study population. There was no statistically significant difference in the proportion of surgical procedures across genders. The rural residents were 45% less likely to be hospitalized compared to their urban counterparts. Among a list of health compromising behaviors, only alcohol consumption was significantly associated with hospital admissions. The history of reporting hospital admission was 64% and 42% lower among former alcohol consumers and those who never consumed alcohol compared to current consumers. Moderate to severe pain was significantly associated with hospital admissions and was 8.8 times more likely to be admitted to the hospital compared to individuals with mild pain. The presence of additional spinal pain was significantly associated with hospital admission and 1.46 times more likely to report a history of hospital admission compared to individuals who had no additional spinal pain. Sciatica was also significantly associated with a history of hospital admissions.

Major-Helsloot et al. investigated the current management strategies for any type of LBP perceived by the patient of the public PHC facilities in Cape Town, South Africa [32]. The majority (90%) of the participants indicated that pain medication was the only form of treatment received, 33% using two or more prescribed painkillers at the time of the study. Ninety-three percent (93%) of the participants with CLBP received pain medication as the only form of treatment, while 15.9% of the sample with LBP received physiotherapy treatment which were mostly, massage, exercises and hot packs (heat therapy). Most participants (76.7%) indicated that, there was not any form of education on predisposing factors offered at their CHC, with only a minority (23.3%) confirming that they were educated on that. Among those that were currently employed at the time of the study, 85% of them reported that they have never received any education or advice on ergonomics to prevent LBP. In terms of satisfaction and the effectiveness of treatment, 36.6% reported that treatment had not helped at all, 48.2% indicated that treatment helped for a short while and only 15.2% responded that treatment helped significantly.

### 3.6. Regional and Gender-Based Differences

Significant regional gender-based differences were noted in terms of both prevalence and risk factors (Table 5). In west Africa, uneducated females had a higher (36.2%) prevalence of CLBP compared to their male counterparts (29.0%) [27]. Chronic LBP was also found to be more prevalent among professional females in technical skills compared to their male counterparts in west Africa [27]. Single males (which included widowed, separated and those that were never married) had a higher CLBP prevalence as compared to those that were married. Oppositely, married females had a higher CLBP prevalence compared to single females [27]. Similarly, in Southern Africa, females demonstrated a higher prevalence of CLBP to males, which ranged from 19.8% to 23.9% among females and 15.9% to 19.7% among males [28,29,34]. A similar finding was observed in central Africa, in which CLBP is more prevalent among the female gender [33]. The incidence of CLBP was also reported to be higher among females in west Africa than among males [25,26]. The etiology of CLBP among west African females was mainly spondylolysis, sacroiliac joint strain, and back strain while disc prolapse was the main cause of CLBP among males [35]. The rate of hospitalization due to LBP was higher among females in east Africa compared to males [35]. The risk factors of CLBP varied considerably among populations and occupation groups due to differences in environmental predispositions. The most commonly identified risk factors were a mixture of both biomechanical and psychosocial factor. Blue collar occupations that involved a lot of lifting heaving objects with repetitive bending and twisting movements, such as farming [26] and driving [28], were found to be more risky for the development of CLBP, whereas white collar occupations were mostly associated with the psychosocial risk factors [1,24].

## 4. Discussion

The purpose of this review was to map the evidence on the distribution of CLBP among in Sub-Saharan Africa with estimates on prevalence, incidence, risk factors, comorbidities and disability. There is variability in the results retrieved in this review due to the differences in the settings of the studies (population-based, hospital-based, occupation-based), and study populations. Some studies included only participants with LBP, which means the high prevalence of CLBP would be expected, while others recruited participants with and without LBP and a lesser prevalence of CLBP would be expected. Hospital-based studies involved participants presenting to the health facility for health-related services (LBP or other ailments). This implies that, these participants had other health problems (comorbidities) which could potentially inflate the prevalence of CLBP. Chronic low back pain is mostly described as a subheading or in a few sentences in studies investigating LBP or other musculoskeletal conditions. Therefore, the comparability of the included studies is difficulty, thus, further research is needed in this context to unpack more evidence related to the epidemiology of CLBP.

According to this review the prevalence of CLBP among the general public ranged between 18.1% to 28.2% and the average CLBP prevalence among LBP patients ranged from 22.2% to59.1%. This concurs with a systematic review by Meucci et al. reported a CLBP prevalence of 25.4% among Brazilian adults [5]. The result of this review shows that the prevalence of CLBP in SSA is comparable to the prevalence estimates in HICs. A Norwayan study reported the CLBP prevalence of 23.6% among adults [36]. The prevalence of CLBP among adults in Iran was 27.18% [37], and 22.3% among Canadian adults [38]. The result of this review also concurs with global estimates (23.0%) as reported by Balagué et al. [39]. On a contrary, Johannes et al. reported a CLBP prevalence rate of 8.1% among US adults [40]. This observation was considerably lower than the present review. Another study by Jiménez-Sánchez et al. reported CLBP prevalence of 14.5% and 7.8% among Spanish females and males, respectively [41]. The prevalence of CLBP in UK is estimated at 11.1% [42]. These observed low prevalence rates of CLBP in some HICs can be attributed to their advancement in the healthcare systems (including equal access to quality health care, health insurance coverage and increased level of health awareness), legislation (such as the no lifting policy) and digitalization of production systems which reduces exposure to occupational biomechanical risk factors.

According to the results of this review, the prevalence of CLBP varied considerably, potentially due to differences in study methodologies employed including variations in study populations, designs, sample sizes and definition of CLBP. The results of this review show that, according to Kebede et al. the prevalence of CLBP among teachers is 30.1%. This result is comparably lower than what was observed by Claus et al. in a similar study conducted in Germany, which reported a CLBP prevalence of 38.7% among school teachers [43]. The differences of which can be attributed to the differences in the male to female ratios in these two studies. Claus reported that 86.8% of the respondents were females, whereas in a study by Kedebe et al. only 54.2% of the participants were females [1]. The other reason for the differences could be attributed to the differences in the willingness to report symptoms between high-income and low-and-middle-income countries [4]. Evidence has shown that there is a great awareness of LBP in high-income countries and people are willing to report symptoms as compared to low-and-middle-income countries [4,44].

In this review, the prevalence of CLBP among farmers was 48.5%. This is in line with what was reported in Thailand, who reported that the prevalence of CLBP was 46.3% among rice farmers [45]. The prevalence of CLBP among Brazilian tobacco farmers was 8.4% with no significant difference between males (7.8%) and females (9.3%) [13]. This high prevalence of CLBP in LMICs can be due to the practice of subsistence farming with long exposure to manual physical work with limited or zero access to machinery. The practice of subsistence farming involves frequent bending and twisting movements, awkward postures, continual unnatural and heavy loading involving lifting, pulling and pushing too heavy causing repeated strain on the back muscles [46,47]. This study also reported that the prevalence of CLBP was higher among office workers than among farmers. This concurs with the postural theorem as was described in detail by Mafuyai et al. [48]. Efforts should be directed towards the development of prevention guidelines and measure should be instigated to ensure adherence. Community stakeholders are recommended to ensure equal access and availability of state-owned farming machinery in underserved communities, which can be shared among community members in order to prevent farmers from indulging in high-risk activities.

The current review found that the prevalence of CLBP among truck drivers was 55.3%. Similarly, a study in Denmark reported the prevalence of CLBP was 57% among bus drivers [49]. These results are consistent with other observations among drivers and official workers [50,51,52]. However, a higher (73.0%) of CLBP among public bus drivers in India in 2016 [53]. This variation could be due to the differences in methodology, road infrastructure and study sample sizes. The study in India which reported a high prevalence of CLBP was estimated with a high sample size. The high prevalence of CLBP among drivers is potentially due to longer duration of improper sitting posture (poor ergonometric), whole body vibration, poor road infrastructure especially in LMICs, lack of exercise and poor nutrition [54,55].

Chronic low back pain is considered a multifactorial condition with multiple etiologies. The risk factors of CLBP identified in this review (SSA) are similar to those reported in high-income countries. Unlike the traditional ideas of biomechanical or biomedical causes of LBP, this review agrees with other studies that have appreciated the biopsychosocial model of etiology. This model (biopsychosocial) acknowledges that physical/biomechanical [1,28,36], psychological [24], socioeconomic [24,27], individual/lifestyle and demographic factors [27] are all equally important predisposing factors for CLBP.

This review has identified poor postural ergonomics, prolonged standing, prolonged sitting, whole body vibration, repetitive bending and twisting movements and lifting heavy objects as the most important biomechanical risk factors for CLBP. This result is in line with reviews from high income countries [56,57,58]. In order to mitigate the risk of CLBP due to occupational biomechanical stressors, proper workplace risk assessment by a qualified professional should be made mandatory and measures should be taken to ensure adherence to proper office ergonomic set-ups, no lifting policy, health education and promotion at workplace should be implemented. Policy makers, stakeholders, funders and other involved actors should ensure equal access to affordable quality health care.

This review acknowledges the role of psychosocial factors in the prediction of chronicity among LBP patients and these factors include fear-avoidance beliefs, abnormal illness behavior, stress, anxiety and depressive disorders. This concurs with studies in high income countries which have reported that early identification of psychosocial risk factors is important in minimizing the risk of progression to CLBP which is associated with greater disability, decreased quality of life and high economic costs [59,60,61,62,63,64]. A UK study has developed a risk assessment tool (STarT Back) for use in primary care to screen for yellow flags or psychosocial risk factors for CLBP [65,66]. This tool has been recently adapted and validated for use in South Africa [67]. However, evidence suggest that the management of LBP in primary health care do not conform to the current guidelines but still comply with the traditional biomedical approach which has already been proven to be ineffective. Thus, measures should be taken to ensure guideline adherence such as regular monitoring and evaluation programs.

Socioeconomic and individual or lifestyle factors such as low socioeconomic status, no formal education, lack of social or family support, excessive or chronic alcohol consumption, cigarette smoking, overweight/obesity, lack of regular physical exercises have also been shown to be important risk factors for CLBP. This finding is also in line with reports from high income countries [68,69]. Ensuring equal access to quality affordable education is a basic human right and urgent action should be taken to ensure that is implemented especially in underserved rural communities. Health education and promotion has been putting emphasis on the HIV/AIDS programs, tuberculosis, cancer, hypertension, diabetes mellitus and other highly fatal diseases, with little attention placed on CLBP, the leading driver of disability. Ensuring limited access to tobacco can help to reduce the risk of poor progression. As the major cause of disability, CLBP should become a priority in terms of research and funding, forming part of the strategy in the implementation of the national development goals.

This is an important review to undertake in this context. Low back pain has been extensively investigated, with little attention placed on the chronic sub-category of LBP. Low back pain is considered a trivial condition because of its smaller association with mortality, however, CLBP is significantly associated with greater disability which affects quality of life. The sustainable development goals (SDGs) aim to reduce the burden of disease and improves quality of life in SSA, therefore, the distribution of CLBP, a leading driver of disability, is important to understand in order to ensure sufficient allocation of resources, guide in the development of context-specific prevention and management guidelines and policy implementation strategies. Considering a great variability of studies due to methodological differences, case definitions and study populations, there is a need to standardize CLBP research methods and case definition to ensure an easy comparability of studies.

## 5. Conclusions

This review has shown that the prevalence of CLBP in SSA is high and comparable to estimates reported in HICs. We also concluded that CLBP is a condition of multifactorial etiologies and requires a holistic multidisciplinary biopsychosocial approach to management. The risk factors for CLBP as identified by the current review encompass physical/biomechanical stressors, psychological, socioeconomic, individual/lifestyle and demographic factors. Although, numerous guidelines have been published in HICs, the diverse differences in culture as regards pain perception and beliefs, nature and organization of the health care systems calls the need for local data in SSA to guide development of culturally validated patient-centered cost-effective management guidelines. Therefore, the extrapolation of results of studies from HICs into the SSA context is difficulty, thus more research effort is needed to close these knowledge gaps in SSA.

## Figures and Tables

**Figure 1 ijerph-19-02964-f001:**
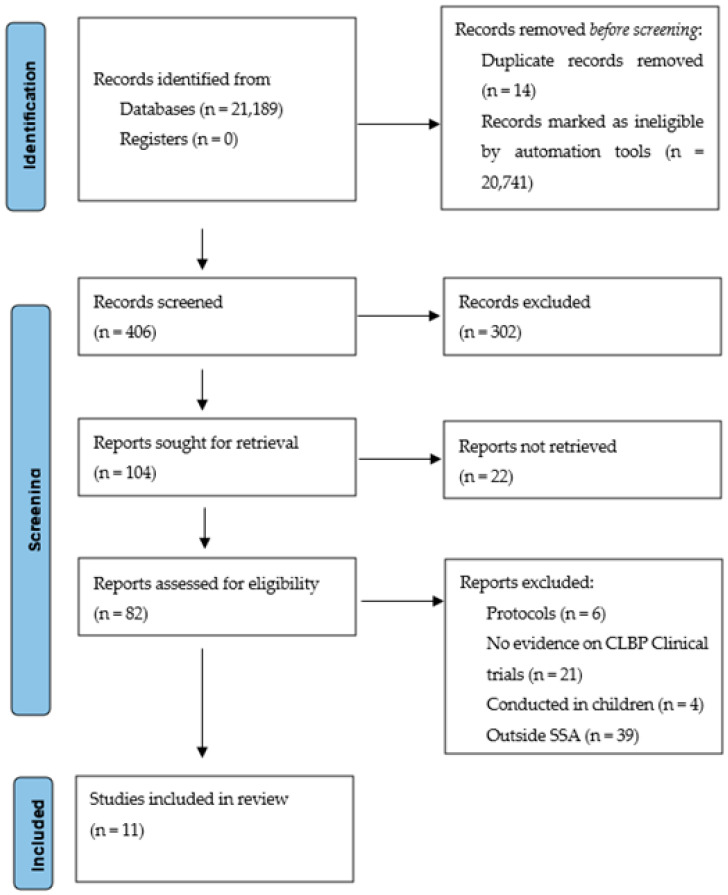
The PRISMA Flow diagram.

**Table 1 ijerph-19-02964-t001:** Data extraction form.

Author and Publication Year	Country	Study Setting	Design	Sample Size	Population Description	Age	Main Findings
Igwesi-Chidobe et al. [24]	Nigeria	Population-based	Cross-sectional	N = 200 Female = 112 Male = 88	General population with low back pain	48.6 (12.0) years	*Risk factors for CLBP disability* Abnormal illness perceptions, severe pain intensity, catastrophizing, FAB, anxiety, lack of social support and female gender
Omoke et al. [25]	Nigeria	Hospital-based	Cross-sectional	N = 291 Female_n_ = 143 Male_n_ = 148	General population with low back pain	45.8 ± 1.67 years	*Prevalence of CLBP * (59.1%) *Incidence* M = 58.3% F = 41.7% *Risk factors* Heavy lifting, previous back injury, obesity, pregnancy, long-distance driving *Comorbidities* Hypertension, diabetes mellitus, peptic ulcer disease, hip and knee osteoarthritis, cervical spondylosis
Tella et al. [26]	Nigeria	Population based	Cross-sectional	N = 604 Female_n_ = 236 Male_n_ = 368	Farmers		*CLBP prevalence* (48.5%) *Risk factors* Prolonged bending
Nakua et al. [27]	Ghana	Population-based	Cross-sectional	N = 4724 Rural: [N = 2799, Women = 1333 Men = 1466] Urban: [N = 1925, Women = 1 Men = 881]	General population	Rural: Female = 64.9 (95%CI: 64.4, 65.5) Male = 64.1 (95%CI: 63.6, 64.7)	*CLBP prevalence* (28.2%) Residence (rural and urban) did not appear to influence the prevalence of chronic back pain
Kebede et al. [1]	Ethiopia	Primary schools	Cross-sectional	N = 611 Female_n_ = 331 Male_n_ = 280	Teachers	40 (±9.38) years	*CLBP prevalence* (30.1%) *Risk factors* sleeping disturbance, prolonged standing, physical inactivity
Yosef et al. [28]	Ethiopia	Port-based	Cross-sectional	N = 422	Truck drivers	37.7 (±9.13 SD) years	*CLBP prevalence* (52.37%) *Risk factors* cigarette smoking, physical inactivity, comorbid chronic diseases, heavy lifting, perceived improper sitting posture, perceived job stress
Beyera et al. [29]	Ethiopia	Population-based	Cross-sectional	N = 543 Female_n_ = 227 Male_n_ = 316	General population with low back pain	43 years (interquartile range 33–55 years)	*CLBP prevalence* Lasting 1–5 years (38.7%), Lasting > 5 years (16.2%) *Factors associated with admissions* Female gender, advanced age, low socioeconomic status, alcohol consumption, severe pain intensity, presence of additional spinal pain
Kahere et al. [30]	South Africa	Hospital-based	Cross-sectional	N = 678 Female_n_ = 394 Male_n_ = 284	General population		*CLBP prevalence* (18.1%) *Risk factors* overweight, no formal education, lack of regular physical exercises, cigarette smoking, alcohol consumption, sedentary lifestyle, manual work, stooped posture
Kahere et al. [31]	South Africa	Hospital-based	Cross-sectional	N = 554 Female_n_ = 228 Male_n_ = 326	General population		*CLBP prevalence* (22.2%) *Risk factors* female gender, middle aged adults 38–47 years, obesity, disease conviction, affective disturbance, denial, FAB
Major-Helsloot et al. [32]	South Africa	Health facility based	Cross-sectional	N = 504 Famale_n_ = 374 Male_n_ = 130	General population	44.8 (SD ± 13.95)	*CLBP prevalence* (26.3%) *Management* Pain medication was the only form of treatment received *Risk factors* psychosocial distress
Doualla et al. [33]	Cameroon	Hospital-based	Cross-sectional	N = 136 Female_n_ = 87 Male_n_ = 48	General population with low back pain	50.6 ± 12.2	*Prevalence of CLBP disability *(88.1%) *Factors associated with greater CLBP disability* pain intensity, longer days of work absence, BBDS *Factors associated with lesser CLBP disability* alcohol consumption, higher psychological wellbeing scores

Note: This table includes all the articles retrieved in the final search.

**Table 2 ijerph-19-02964-t002:** Methodological quality assessment of included studies.

Score Weight	0.2	0.08	0.2	0.2	0.08	0.08	0.08	0.08	Total Score
Study ID	Was the Sampling Frame a True or Close Representation of the Target Population	Was the Sample Size Estimated?	Was Some Form of Random Selection Used to Select Sample or Was a Census Undertaken?	Was the Likelihood of Nonresponse Bias Minimal?	Were Data Collected Directly from the Subjects (as Opposed to a Proxy)?	Had the Study Instrument That Measured the Parameter of Interest (e.g., CLBP Prevalence) Been Tested for Reliability and Validity?	Was Data Collection Standardized?	Was a Human Body Drawing Used	
Igwesi-Chidobe et al. [24]	Yes	No	Yes	Yes	Yes	Yes	Yes	No	0.84
Omoke et al. [25]	Yes	No	No	Yes	Yes	Yes	Yes	No	0.64
Tella et al. [26]	Yes	No	No	No	Yes	Yes	Yes	No	0.44
Nakua et al. [27]	Yes	No	Yes	Yes	Yes	Yes	Yes	No	0.84
Kedebe et al. [1]	Yes	Yes	Yes	Yes	Yes	Yes	Yes	No	0.92
Yosef et al. [28]	Yes	Yes	Yes	Yes	Yes	Yes	Yes	No	0.92
Beyera et al. [29]	Yes	Yes	Yes	Yes	Yes	Yes	Yes	No	0.92
Kahere et al. [30]	Yes	Yes	Yes	Yes	Yes	Yes	Yes	No	0.92
Kahere et al. [31]	Yes	Yes	Yes	Yes	Yes	Yes	Yes	No	0.92
Major-Helsloot et al. [32]	Yes	Yes	Yes	Yes	Yes	Yes	Yes	No	0.92
Doualla et al. [33]	Yes	No	Ye	No	Yes	Yes	Yes	Yes	0.84

**Table 3 ijerph-19-02964-t003:** Characteristics of included studies.

Author and Publication Year	Country	Study Setting	Design	Sampling Method	Sample Size	Population Description	Age
Igwesi-Chidobe et al. [24]	Nigeria	Population-based	Cross-sectional	Random	N = 200 Female = 112 Male = 88	General population	48.6 (12.0) years
Omoke et al. [25]	Nigeria	Hospital-based	Cross-sectional	NR	N = 291 Female_n_ = 143 Male_n_= 148	General population	45.8 ± 1.67 years
Tella et al. [26]	Nigeria	Population based	Cross-sectional	NR	N = 604 Female_n_ = 236 Male_n_ = 368	Farmers	NR
Nakua et al. [27]	Ghana	Population-based	Cross-sectional	NR	N = 4724 Rural: [N = 2799, Women = 1333 Men = 1466] Urban: [N = 1925, Women = 1 Men = 881]	General population	Rural: Female = 64.9 (95%CI: 64.4, 65.5) Male = 64.1 (95%CI: 63.6, 64.7)
Kahere et al. [30]	South Africa	Hospital-based	Cross-sectional	Random	N = 678 Female_n_ = 394 Male_n_ = 284	General population	NR
Kahere et al. [31]	South Africa	Hospital-based	Cross-sectional	Random	N = 554 Female_n_ = 326 Male_n_ = 228	General population	45.8 ± 10.7
Major-Helsloot et al. [32]	South Africa	Health facility based	Cross-sectional	Random	N = 504 Famale_n_ = 374 Male_n_ = 130	General population	44.8 (SD ± 13.95)
Kebede et al. [1]	Ethiopia	Primary schools	Cross-sectional	Random	N = 611 Female_n_ = 331 Male_n_ = 280	Teachers	40 (±9.38) years
Yosef et al. [28]	Ethiopia	Port-based	Cross-sectional	Random	N = 422	Truck drivers	37.7 (±9.13 SD) years
Beyera et al. [29]	Ethiopia	Population-based	Cross-sectional	Random	N = 543 Female_n_ = 227 Male_n_ = 316	General population	43 years (interquartile range 33–55 years)
Doualla et al. [33]	Cameroon	Hospital-based	Cross-sectional	Random	N = 136 Female_n_ = 87 Male_n_ = 48	General population	50.6 ± 12.2

**Table 4 ijerph-19-02964-t004:** Prevalence of chronic low back pain.

Prevalence of CLBP among Occupations				
Author	Year	Country	Occupation	Prevalence of CLBP
Kebede et al.	2019	Ethiopia	Primary school teachers	30.1%
Tella et al.	2013	Nigeria	Farmers	48.5%
Yosef et al.	2019	Nigeria	Truck drivers	55.5%
**Prevalence of CLBP among LBP Patients**				
Author	Year	Country	Population	Prevalence of CLBP
Omoke et al.	2016	Nigeria	LBP patients	59.1%
Beyera et al.	2020	Ethiopia	LBP patients	38.7% for >1 year
Doualla et al.	2019	Cameroon	LBP patients	41.0%
Kahere et al.	2022	South Africa	LBP patients	22.2%
**Prevalence of CLBP among the General Population**				
Nakua et al.	2015	Ghana	General population	28.2%
Major-Helsloot et al.	2014	S. Africa	General population	26.3%
Kahere et al.	2021	S. Africa	General population	18.1%

**Table 5 ijerph-19-02964-t005:** Regional and gender-based analysis.

SSA Region	Countries	No. of Studies	Outcome Measure (s)	Main Findings	Gender-Based Differences
West Africa	Nigeria, Ghana	4	CLBP Prevalence Risk factors CLBP disability Etiology Pattern of presentation CLBP burden	*Prevalence of:* CLBP—28.2–48.5% Self-reported CLBP disability—62.5% Performance-based disability—49.1% *Risk factors of:* *CLBP—*lifting heavy objects, history of back injury, obesity, pregnancy, long distance driving, prolonged bending *Self-reported CLBP disability*—abnormal illness perception, severe pain intensity, catastrophizing, fear avoidance beliefs, anxiety *Performance-based disability*—abnormal illness perception, lack of social support, fear avoidance beliefs, female gender *Comorbidities*—hypertension, peptic ulcers, diabetes mellitus, knee and hip osteoarthritis, cervical spondylosis *Etiology* Lumbosacral spondylosis, degenerative spondylolisthesis,	*Prevalence of CLBP* F_(uneducated)_ (36.2%) > M_(uneducated)_ (29.0%) F_(professionals)_ (40.8%) > M_(professionals)_ (28.0%) M_(single)_ > M_(married)_ F_(married)_ > M_(single)_ *Incidence* F (53.1%) > M (46.9%) *Etiology* *Females—*spondylosis, spondylolisthesis, sacroiliac joint strain and back strain *Males—*disc prolapse
East Africa	Ethiopia	3	CLBP Prevalence Admission rate Factors associated with healthcare presentation for LBP	*Prevalence* 30.1% *Risk factors* Smoking, physical inactivity, chronic diseases, frequent lifting and carrying heavy objects *Admission rate* 14.4% (95%CI: 11.4–17.3) *Factors associated with LBP presentation* Lack of social or family support (living alone) Alcohol consumption Severe pain Presence of additional spinal pain	*Prevalence*—no gender stratified analysis *Risk factors—*no gender stratified analysis *Admission rate* F (18.5%) > M (11.4%)
Southern Africa	South Africa	3	CLBP prevalence Risk factors Management	*Prevalence* 18.1–26.3 *Risk factors* *Lifestyle*: (Overweight, smoking, alcohol consumption, lack of regular physical exercise) *Occupational*: (prolonged sitting, manual work, stooped posture) *Socioeconomic*: (illiteracy) *Psychological*: (disease conviction, affective disturbance, denial, fear avoidance beliefs about work) *Management* Biomedically oriented, mainly pain medication, rare physiotherapy referral, no advice given, low level of patient satisfaction	*Prevalence* F (19.8—23.9) M (15.9—19.7) *Risk Factors* Female gander was a significant risk factor for CLBP (aOR = 12.4; 95%CI: 3.1–49.8; *p*-value < 0.001) *Management* Management was unrelated to the type of gender
Central Africa	Cameroon	1	CLBP disability	*Risk factors* *Greater disability*: (Pain intensity, longer days of work absenteeism, bladder and bowel dysfunction syndrome) *Lesser disability*: (alcohol consumption, psychological well-being)	F (64%); M (36%) No gender stratified analysis performed

Note: F denotes Females; M denotes Males.

## Data Availability

All data generated or analyzed during this study will be included in the published article.

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
