# Peer review of "A Scoping Review on the Epidemiology of Chronic Low Back Pain among Adults in Sub-Saharan Africa"

_ijerph, 2022, doi:10.3390/ijerph19052964_

Round 1

Reviewer 1 Report

Thank you for the submission of your manuscript. There are significant causes for concern in the manuscript presentation ranging from inconsistent reference list formatting, typos, inconsistency in grammar/abbreviations/etc, incorrect presentation of data. 

I feel this paper is of sufficient interest to be published but significant work is needed before it can be considered. I have attached some comments from reviewing up until the results section. 

Author Response

The response to reviewers comments are detailed in the attached file.

Reviewer 2 Report

The systematic review is methodologically well described and structured. In the introduction section it is described under the concept of causes of absenteeism and as a highly disabling pathology.

Line 74: The authors refer that there is no clear definition of the term chronic low back pain in terms of its anatomical characteristics and duration of symptoms. It would provide a definition of low back pain according to the consensus and results obtained in the articles reviewed.

Line 347: The authors speak of physical exercise as a risk factor for low back pain. This should be better specified; perhaps they are referring to poorly controlled exercise, without adjustments of loads, dosage, frequency.

Exercise is a tool widely used in physiotherapy programs to treat low back pain and with very good scientifically proven results.

In the discussion section, would it be possible for the authors to clarify in line 460-462 the variability of the studies in terms of data collection from people with and without low back pain. Should this type of articles be excluded?

Author Response

The response to reviewers comments are details in the attached file.

Reviewer 3 Report

Reviews

  1. Abstract:
  • The abstract is nicely written. However, I recommend what is the outcome of your literature search? and if you can provide this information in conclusion. It will suffice the conclusion content.

  1. Introduction
  • Line 75-80, The introduction needs to focus on why the literature search is performed. Although the purpose of the study is clear, what exactly motivated the researcher to conduct the literature search. This question needs to be addressed.
  • I recommend adding one more paragraph and some more literature that connects the story well with the purpose of your study.

  1. Methods

  • Line 107, What do you mean by grey literature?
  • Why studies including children were excluded from this literature search. Can you provide a rationale for it?
  • Kindly also provide whether the data extraction form table and article included in it is the final result of your search. You can add that as a footnote for your table.

  1. Results

  • I like the way you have presented your result search with table and figures
  • The analysis and commentary on results are nicely presented using tables.
  • I encourage doing the same analysis and presenting your data as a table on a regional basis. I saw that on pages 12-13 you have provided information on certain regions and populations in SSA that have CLBP issues. Can you make a table and provide information on whether there are certain trends, occupation, or other issues and factors that a specific SSA population faces that leads to higher chances of CLBP? Present it as a table. You can also show gender-based differences too in these communities. It will enhance the quality of your article. I also encourage thinking about the classes for your table and organizing so that a reader can just go through the table rather than reading through the text. If he has to look for more information, then he/she/they can tend to your text.

  1. Discussion

  • In discussion, kindly provide the novelty of your review article. Why is your review article is novel and is different than any other article that is in the same scope?
  • The discussion is nicely written I suggest adding a paragraph where you should add about the implication of this article.

  1. Conclusion
  • The conclusion requires more clarity. There are some grammatical issues. I encourage you to write about what is you can draw from this literature that can be beneficial. Moreover, if there are multiple factors associated with CLBP what are the most important/prominent factors that result in CLBP issues in SSA, and what regions are highly impacted by it based on your search results.

Author Response

(The authors gave the same response as above.)

Reviewer 4 Report

Although the topic seems to be interesting, the preparation of the manuscript is far from accepted in the high score impact factor journal, considering both the editorial and substantive issues.

Editorial corrections are necessary for the literature citations in References, some of the refs. are not included in the PubMed database or the titles are wrongly written which makes it impossible to find the source on the journal website.

The state of art in the Introduction section includes refs. looking to be chosen accidentally in some cases. Some of the sentences in the whole manuscript in their meaning include well-known statements fro the handbooks (e.g. lines 16-17 …” Reviews on epidemiological estimates in low-and-middle-income-countries are important to undertake.”… yes, they are).

Finally, authors in their Review have found 10 papers on the topic worth to be discussed, and treat them as the basis for the reliable analysis. Hence the conclusions are generalized and easy to predict.

The inclusion and exclusion criteria are not formulated precisely and perhaps for this reason the study result leads to generalized conclusions.

One of the strengths of the work is the applied PRISMA methodology.

In conclusion, the paper needs significant improvements, since the current form is difficult to be accepted.

Author Response

(The authors gave the same response as above.)

Round 2

Reviewer 4 Report

The paper improved significantly. The literature has been updated. The authors addressed my queries in most of the cases. The editorial improvements are necessary:

-Data in Table one is unclear because of the editorial reason and it is difficult to read because of pdf format. Please provide Word version. Native speaker corrections interfere with the content which is important. Please leave only sentences after revisions.

-PRISMA design in Figure 1 should be left only in the final version without cross-outs. In delivered pdf file is absolutely unclear. 

-Please check again English spelling and abbreviations. E.g. West Africa not west Africa on page 65 and throughout the text.